# GSNB: Gaussian Splatting With Neural Basis Extension

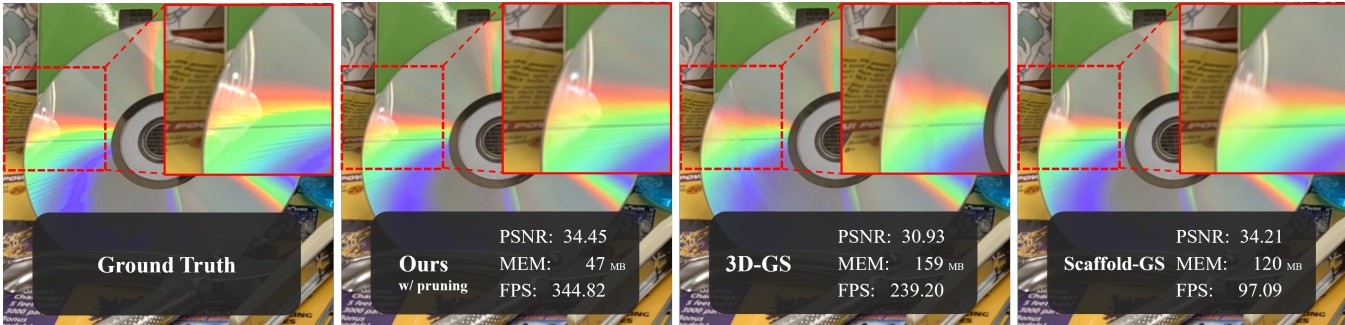

**Figure 1: Our proposed GSNB achieves impressively accurate visual effects and renders high-quality images in real-time. Our progress relies on a Neural Basis Extension module to complement color calculation. This module enables adaptable supplements to spherical harmonic basis functions, facilitating the modeling of intricate visual effects. Additionally, we employ baking technique to precompute the network and devise an importance score for model pruning, further enhancing rendering efficiency.**

## ABSTRACT

The 3D Gaussian Splatting (3D-GS) method has recently sparked a revolution in novel view synthesis with its remarkable visual effects and fast rendering speed. However, its reliance on simple spherical harmonics for color representation leads to subpar performance in complex scenes, particularly with effects like specular highlights and light refraction. Also, 3D-GS adopts a periodic split strategy, which significantly increases the model's disk space and hinders rendering efficiency. To tackle these challenges, we propose Gaussian Splatting with Neural Basis Extension (GSNB), a novel approach that substantially enhances the performance of 3D-GS in demanding scenes while reducing storage consumption. Drawing inspiration from basis function, GSNB utilizes a light-weight MLP to share feature coefficients with Spherical Harmonics (SH). This extends the color calculation of 3D Gaussians, resulting in more accurate visual effect modeling. This combination allows GSNB to achieve remarkable results even in scenes with challenging lighting and reflection conditions. Additionally, GSNB uses pre-computation to bake the MLP's output, thereby alleviating inference workload and subsequent speed loss. Furthermore, to leverage the capabilities of Neural Basis Extension and eliminate redundant Gaussians, we propose a new importance criterion to prune the converged Gaussian model and obtain a more compact representation through re-optimization. Our experimental results demonstrate that our method delivers high-quality rendering in most scenarios and

effectively reduces redundant Gaussians without compromising rendering speed. Our code and real-time demos will be released soon.

## CCS CONCEPTS

• **Computing methodologies → Rendering**; **Appearance and texture representations**.

## KEYWORDS

novel view synthesis, radiance fields, 3D gaussians, real-time rendering

## 1 INTRODUCTION

Novel view synthesis focuses on generating images from new viewpoints using a collection of images captured from a scene. This technology has broad applications in virtual reality, augmented reality and 3D film production. Some approaches extract primitives that can be rapidly rasterized, such as meshes [30] or points [19, 52], facilitating scene reconstruction and swift rendering. In contrast to these explicit methods, Neural Radiance Field (NeRF) [27] uses a neural network to represent the scene's geometry and appearance information. It employs a classical volume rendering process to query the network and delivers high-quality synthesis results. However, the intensive network queries make it challenging to apply to real-time rendering.

Following NeRF, 3D Gaussian Splatting (3D-GS) [16] has gained widespread attention recently for its point-based approach, which achieves state-of-the-art quality and rendering speed. This method represents the scene as a set of 3D Gaussians with anisotropic attributes. By leveraging GPU-accelerated and tile-based differentiable rendering, it swiftly computes gradients corresponding to each Gaussian's attributes, thereby updating the underlying scene representation.

Despite its superb performance, 3D-GS struggles with complex effects such as specular reflection and light refraction, as shown in Figure 4. This limitation stems from the low-degree spherical harmonics used in 3D-GS, which are insufficient for capturing the high-frequency information in such scenes. Additionally, the splitting strategy employed by 3D-GS significantly increases memory consumption, millions of Gaussians place considerable demands on the hardware, impacting rendering efficiency.

To address these challenges, we present Gaussian Splatting with Neural Basis Extension (GSNB). This method combines spherical harmonic functions with a set of neural network-based "basis functions". This integration allows the model to capture rapidly changing phenomena, such as specular highlights, significantly improving the view synthesis performance of 3D-GS in complex scenes. Concurrently, we improve the storage efficiency of the GS representation through an extra pruning process, resulting in a more compact and efficient model. Specifically, GSNB utilizes a lightweight MLP, known as Neural Basis Extension, to capture viewpoint-related colors. The result is then combined with the base color obtained from the spherical harmonics to reproduce fine visual effects. As depicted in Figure 1, GSNB markedly enhances the performance of 3D-GS in scenarios with high-frequency information. Additionally, by baking the network into images that occupy minimal space, GSNB reduces the real-time network inference cost during rendering, which increases linearly with the number of Gaussians, thus ensuring minimal loss in rendering speed. To further leverage the expressive capabilities of Neural Basis Extension and achieve a more compact scene representation, we propose a novel importance score to assess contribution of each Gaussian to the imaging results during training. Then we prune the converged Gaussian model based on this criterion. This approach considerably decreases the model's space consumption with minimal impact on rendering quality.

By combining the above methods, GSNB has made striking progress in visual quality, storage consumption, and rendering speed simultaneously. Moreover, experiments demonstrate that GSNB performs well not only in complex scenes but also achieves state of the art results in general datasets.

In summary, our method makes the following contributions:

- A novel hybrid appearance model for Gaussian Splatting that utilizes a lightweight MLP to extend the color expressed by spherical harmonics and capable of capturing delicate visual effects in complex scenes.
- An efficient baking method for real-time neural network inference that maintains rendering speed within the Gaussian rasterization framework.
- A new Gaussian pruning strategy that speeds up the rendering process while preserving as much valid information as possible, supported by our proposed Neural Basis Extension.

## 2 RELATED WORK

### 2.1 Neural Radiance Fields

Neural rendering has gained widespread attention in recent years due to its obvious advantages in synthesizing photorealistic images from novel views. Neural Radiance Fields (NeRF) [27] achieves state of the art results at the time of proposal by combining volume rendering with the implicit representation of neural networks. The method uses differentiable rendering to learn from a series of scene pictures and stores the information by multilayer perceptron(MLP) in high quality. A large amount of follow-up work has extended the application area of NeRF and further improved the performance of the method, including rendering and geometry quality [1, 2, 26, 37, 44], few-shot reconstruction [5, 23, 50], 3D-aware generation [3, 4, 34], semantic segmentation [39, 43, 53], and pose estimation [32, 36]. Among them, Mip-NeRF360 [2], which uses a non-linear scene parameterization and online distillation technique, achieves state of the art rendering quality and is used as one of the baselines in our experiments.

However, the ray-tracing approach indicates that NeRF requires a network query at each sampling point. Much work has been devoted to improving the efficiency of NeRF. One approach is to bake the scene information stored in the MLP into data structures that can be accessed quickly, such as voxel grids [14], octrees [49], smaller MLPs [11, 31], mesh vertices [7], etc. The other class of methods contains more innovations, often increasing the speed of training and inference at the cost of memory. NSVF [22] models the scene as a voxel radiance field, and the features are obtained by extracting the learnable features on the voxels and then interpolating them, which is a more efficient approach compared to the dense sampling of NeRF. Instant-NGP [28], on the other hand, combines hash encoding with voxel grid, and greatly improves the training and inference speed of NeRF through a customized CUDA implementation, achieving results similar to those of the original NeRF in just a few seconds of training.

Although the above approaches achieve higher rendering quality and faster rendering speed, they do not fundamentally alter the rendering process of NeRF, which involves ray tracing with sample point lookup. Our work is based on 3D-GS, which uses rasterized rendering to significantly increase the rendering speed while guaranteeing high visual quality compared to NeRF.

### 2.2 Differentiable Point-based Rendering

Point-based rendering methods can efficiently represent discontinuous geometry in the scene and render at high speed through rasterization. The differentiable point-based rendering technique [40, 48] has been widely discussed in recent years due to its ability to automatically fit the scene, which can alleviate problems such as artifacts and holes that may occur in traditional point rendering methods and improve the scene quality. Notably, Pulsar [20] implements a fast sphere rasterization pipeline, which inspired 3D-GS to use 3D Gaussian for scene representation and adopt a tile-based rasterization approach.

As a recently proposed point-based rendering method, 3D-GS simultaneously achieves the same or better quality as the best-quality neural radiance fields based approach and real-time rendering performance. Subsequent improvements quickly emerge and expand the application areas of 3D-GS, including performance optimization [25, 45, 51], mesh extraction [6, 12], dynamic scene [42, 46], 3D content generation [21, 38, 47], and virtual human [15, 18, 54]. Notably, Scaffold-GS [25] uses anchor points to place local 3D Gaussians and predicts their attributes during runtime,

thus regularizes the Gaussians distribution and improves the performance in fine-scale details. SuGaR [12] extracts mesh from 3D-GS representations and attaches 3D Gaussians to mesh to further optimize the underlying Gaussians distribution.

While 3D-GS and its subsequent works have achieved state of the art results in common scenes, there is still much room for improvement in special scenes with high-frequency color changes, such as specular highlights. In this work, we introduce a lightweight network that extends the spherical harmonic function to achieve accurate modeling and real-time rendering of complex scenes, substantially improving the rendering efficiency of 3D-GS.

## 2.3 Model Pruning

The primary goal of model pruning is to strike a balance between model performance and space occupied, reduce resource usage by cutting parameters of relatively low importance and maintain the performance of the pruned model as much as possible. Some representative methods include Structured Pruning [24], Soft filter Pruning [13], the Lottery Ticket Hypothesis [10].

Although the aforementioned approaches focus on pruning the neural networks, their ideas can also be applied to 3D-GS as well [8]. To fit the fine geometric and textural content of the scene, 3D-GS adopts a splitting strategy, which leads to a rapid increase in the number of Gaussians. Millions of Gaussians impede a closer improvement in rendering efficiency. In our work, we design the pruning strategy with the granularity of a single Gaussian and make full use of the power of Neural Extension. The strategy effectively reduces the storage occupancy of GSNB and accelerates the rendering speed.

## 3 OVERVIEW

The overview of our method is presented in Figure 2. For a target scene, the input comprises photos from different angles. The COLMAP [35] calibration process is used to obtain the camera pose for each photo and a set of sparse points that provide initial Gaussian parameters. Next, we use the differentiable rendering method of 3D-GS [16] to update the model data, with spherical harmonics as the initial Gaussian color expression. During training, neural basis extension serves as auxiliary expressions to capture complex lighting information, such as specular reflections, with greater precision. This complements the base color captured by traditional spherical harmonics, enabling more accurate color calculations and enhancing the model's performance in complex scenes. With the help of the network, we can further reduce the number of Gaussians by using novel importance indicators after training convergence. Additionally, we can restore the loss of infomation caused by pruning through a re-optimization process.

## 4 GAUSSIAN SPLATTING WITH NEURAL BASIS EXTENSION

Our proposed model, GSNB achieves high-quality novel view synthesis quality in challenging scenes. It primarily consists of a basic set of 3D Gaussians and a Neural Basis Extension module for modeling intricate effects. The model supports high speed real-time rendering by network baking and model pruning, which we will detail as follows.

### 4.1 Gaussian Splatting Representation

The imaging principle of 3D-GS [16] aligns with the volume rendering method employed by NeRF [27]. However, a key difference is that NeRF retrieves the volume density and color of each sampling point along the ray using an MLP network. This approach results in an inevitable network query that slows down the rendering process.

To address this issue, point-based rendering replaces the sample points with explicit points and calculates the color through Eq. (1). Each ray considers only the corresponding alpha and appearance of the points to acquire the color.

$$C = \sum_{i \in N} c_i \alpha_i \prod_{j=1}^{i-1} (1 - \alpha_j) \tag{1}$$

Furthermore, 3D-GS represents these points as 3D Gaussian function that can be updated in scale and rotation, further increasing the flexibility of point-based rendering. Specifically, the Gaussian function in 3D space can be defined by Eq. (2), where $\Sigma$ denotes the covariance matrix of this Gaussian function.

$$G(x) = \exp(-\frac{1}{2}(x)^T \Sigma^{-1} (x)) \tag{2}$$

On the other hand, the color of each 3D Gaussian is represented by spherical harmonics. For each color channel, 3D-GS uses this basis function to match view-dependent colors in the scene. The degree of the spherical harmonics is gradually increased during the training process to simulate color variations from coarse to fine. However, fixed spherical harmonics struggle to express complex high-frequency color changes especially specular highlights. To accurately represent such difficult scenarios using spherical harmonics, the required degree of the basis functions must be increased, resulting in increased space consumption and training difficulty.

To address this challenge, we propose a solution that complements the spherical harmonics with a lightweight MLP, which we call Neural Basis Extension, to capture view-dependent information. This approach provides a more effective representation for scenarios where fixed basis functions fall short and allows us to capture subtle color variations in the scene, resulting in improved rendering quality.

### 4.2 Neural Basis Extension

The core thinking of GSNB is to construct neural basis functions defined by an MLP network parallel to the original 3D-GS spherical harmonics and produce a hybrid appearance model.

For a 3D Gaussian $G$ in original 3D-GS [16], the color of each channel can be modeled as a function $\mathbb{C}^G(v) : R^3 \rightarrow R$ with respect to a direction vector $v : (v_x, v_y, v_z)$ as follows.

$$\mathbb{C}^G(v) = \sum_{n=1}^{N} k_n^{G_{sh}} SH_n(v) , \tag{3}$$

where $SH_n(v), n = 1, \ldots, N$ are $N$ spherical harmonic basis functions, and $k_n^{G_{sh}}$ are the corresponding SH coefficients. In this paper, $N = 16$ for the first three degrees of SH.

However, to express the high-frequency effect in a small range, such as specular highlight, the spherical harmonic function requires a large number of coefficients to incorporate higher-degree terms.

**Figure 2: Framework. Our pipeline begins with a set of images and the point cloud extracted by COLMAP [35]. For the initial iteration, we use a warm-up phase to learn the low frequency color distribution through spherical harmonics without optimizing the neural network. After that, we introduce Neural Basis Extension to capture complex lighting information, serving as a supplement to the colors obtained from spherical harmonics and jointly optimize the 3D Gaussians parameter and the network. Finally, we conduct an extra pruning process to reduce redundant Gaussians.**

This not only significantly increases the model's space requirements, but also makes learning these parameters more difficult. While it is feasible to regress this function directly through a neural network, this NeRF-like approach is very inefficient for real-time rendering. We take inspiration from the form of spherical harmonic basis and use a lightweight MLP network $G_\phi$ with parameters $\phi$, namely Neural Basis Extension (NBE), to define $M$ learnable neural basis functions $NB_n(v), n = 1, \ldots, M$:

$$G_\phi : v \rightarrow (NB_1, NB_2, ..., NB_M) \quad (4)$$

Similarly, we introduce a set of coefficients $k_n^{G_{nbe}}$ for the neural basis functions $NB_n(v), n = 1, \ldots, M$. For the spherical harmonic functions and the neural basis functions, we use the same number of coefficients, i.e., $M = N$. The color from a given viewpoint $v$ for each 3D Gaussian is then calculated as follows:

$$\mathbb{C}(v) = \sum_{n=1}^{N} k_n^{G_{sh}} SH_n(v) + \sum_{n=1}^{N} k_n^{G_{nbe}} NB_n(v) \quad (5)$$

Nevertheless, using two different sets of coefficients simultaneously takes up a lot of extra space. In our experiments, we use a coefficient sharing method, that is, the spherical harmonics and NBE share the same set of basis function coefficients. For training stability, we first perform a warm-up phase in the early stages of training and train the model alone without introducing the neural network. The color at this time is obtained from Eq. (3) to obtain a reasonable initial value. After 3k iterations, we add NBE to the rendering process and use Eq. (5) to calculate the color. Note that at this point $k_n^{G_{sh}}$ and $k_n^{G_{nbe}}$ represent the same coefficient. With

this approach, we can halve the number of coefficients required, resulting in significant space savings without any noticeable loss in quality. Our experiments show that when sharing coefficients, spherical harmonics mainly use low-degree coefficients to represent the basic color of the scene, while NBE is responsible for expressing more complex visual effects. The two cooporate very well on the same set of coefficients.

For the Neural Basis Extension network, we use the difference between the 3D Gaussian center position and the camera position as the input vector. We perform positional encoding on the normalized input values $p$ using the following formula, in all of our test scenes and set $L = 6$.

$$\gamma(p) = (\sin(2^k \pi p), \cos(2^k \pi p))_{k=0}^{L-1} \quad (6)$$

## 4.3 Annealing Perturbation Against Overfitting

During the training process of NBE, the camera poses of the training images are fixed, consequently, for each 3D Gaussian the direction required for color query is also fixed. As a result, the NBE model can develop an inclination to overfit at specific given angles, which lead to abnormal color banding effects and decreased performance in simple scenes. To address the issue, we introduce a perturbation strategy that decays during the training process to mitigate the fixed direction. This strategy adds random noise to the direction vector (input of the network) during training to enhance the network's robustness and ensure more reasonable and continuous output across viewing angle.

This random perturbation in direction can be expressed by as follows:

$$v' = v + \Delta v(i)$$

$$\Delta v(i) = \mathcal{N}(0, I) \cdot \alpha \cdot (1 - \frac{i}{\tau}) \tag{7}$$

Here, $\Delta v(i)$ indicates the linearly attenuated noise added to the input direction vector $v$ in the $i$-th training iteration. $\mathcal{N}(0, I)$ denotes a Gaussian distribution with zero vector as the mean and $\mathbf{I}$ as the covariance matrix. The term $\alpha$ is an empirically determined scaling factor adjusting noise amplitude and is set to 0.3 during training. The variable $\tau$ indicates the total number of training iterations (empirically set to 30k).

Our experiments demonstrate that this strategy allows the model to maintain a smooth color transition across different viewing angles by transforming specific direction input into a vague range. This mitigates abrupt change and unnatural color jump in the rendering results and helps reduce overfitting. By preventing the model from overfitting to specific views in the training data, this perturbation enhances NBE's robustness to generalize to unknown views.

## 4.4 Bakery Based Real-Time Rendering

During rendering, the color of each 3D Gaussian is determined by calculating the value of the spherical harmonics function in real-time. However, when using Neural Basis Extension expressed by MLP, real-time inference of the network can significantly impact performance as NBE has higher computational complexity than spherical harmonics, making it challenging to maintain efficiency in real-time rendering. Therefore, it is essential to consider additional optimization methods to accelerate NBE inference.

We note that the Neural Basis Extension can be viewed as a function that takes a 2D direction vector $v = (\phi, \theta)$ and outputs values over 16 channels. Thus, we precompute the trained MLP and store the result in 16 single-channel images. Each pixel in an image serves as an NBE output value in a particular direction. These baked images are stored in the GPU along with the Gaussian parameters for real-time query. During color computation for each 3D Gaussian, we look up these images based on the current target direction of the Gaussian. This strategy eliminates the need for MLP inference during rendering, significantly reducing the per-frame overhead while only marginally increasing the memory footprint (by about 2 MB in our implementation). Our experiments show that this baking method can achieve high-speed real-time rendering with minimal impact on rendering quality.

## 4.5 Gaussian Pruning With Network Co-adaptation

The 3D-GS implementation uses a periodic splitting strategy based on the gradient of the Gaussian to handle complex scenes. This strategy improves the model's expressive power but also requires significant storage space due to the large number of 3D Gaussians needed to fit a single scene. Each Gaussian requires storage of its position, covariance matrix, and spherical harmonics coefficient. Our approach integrates Neural Basis Extension and uses baked images to facilitate real-time queries of NBE, which consumes additional memory for acceleration purposes.

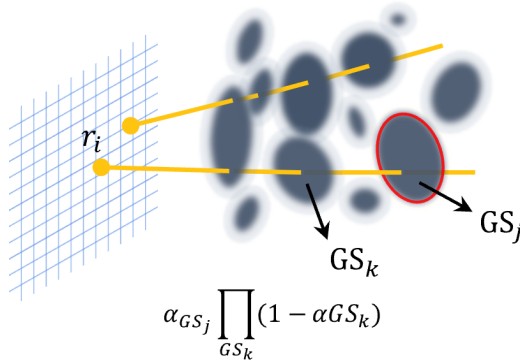

Figure 3: Gaussian importance score. For a Gaussian $GS_j$, we consider a ray $r_i$ that intersects it, that is, $\mathbb{I}(GS_j, r_i) = 1$. Then we calculate the weight of $GS_j$ during the color calculation of $r_i$, which should take into account other Gaussians like $GS_k$ that occlude $GS_j$.

To address this issue, we introduce a new Gaussian importance score. This score assesses the contribution of each Gaussian to every pixel in the training images and then aggregates these values to determine the cumulative contribution of each Gaussian to the entire training dataset. This score will be used to prune the Gaussians, retaining those with larger contributions and eliminating those with smaller ones. The score is calculated using the following formula:

$$I_{GS_j} = \sum_{r_i}^{T \times H \times W} \mathbb{I}(GS_j, r_i)\alpha_{GS_j} \prod_{GS_k}(1 - \alpha_{GS_k}) \tag{8}$$

The above formula determines whether the calculation process of each ray $r_i$ in $T$ training images uses the 3D Gaussian $GS_j$. $H$ and $W$ refer to the image height and width, respectively. If it is used, we consider the weight of the current Gaussian on $r_i$'s final color. $GS_k$ indicates all Gaussians that have an occlusion effect on $GS_j$, and refer to Eq. (1), the weight can be expressed as the product of the alpha value of $GS_j$ and the transmittance calculated using $GS_k$.

After pruning the Gaussians, the model needs to be re-optimized to restore the lost information. This is done by optimizing the parameters of the Gaussian model and the Neural Basis Extension together, which is similar to the prior training process. Our NBE shows more potential to recover the scene compared to fixed spherical harmonics during this joint optimization, allowing us to maintain superior rendering quality even after pruning. Through this mechanism, users can control the number of Gaussians based on their requirements and balance memory consumption with model performance.

## 5 EXPERIMENTS

### 5.1 Experimental Setting

*5.1.1 Datasets and Metrics.* To evaluate the performance of novel view synthesis in static Scenes, our experiments are performed on five widely used datasets: Shiny [41], Spaces [9], Mip-NeRF360 [2], Tanks and Temples [17], and Synthetic NeRF [27]. The Shiny [41]

dataset contains several real-world scenes with complex light and shadow variations, including specular highlights, refraction, and reflection. The Spaces [9] dataset includes numerous image sequences of indoor spaces, capturing geometric details and lighting effects of different materials. Mip-NeRF360 [2] and Tanks and Temples [17] are widely used view synthesis datasets that provide detailed indoor scenes and a variety of outdoor scenes, such as natural landscapes and cityscapes. Synthetic NeRF [27] consists of 8 synthetic image sets for individual objects. Our experimental dataset covers a wide range of scenarios, from indoor to outdoor and from simple to complex scenes.

We apply the three widely-used metrics for evaluation, *i.e.*, peak signal-tonoise ratio (PSNR), structural similarity index measure (SSIM), and the learned perceptual image patch similarity (LPIPS). We additionally report the storage size (MB) and the rendering speed (FPS) for model compactness and performance efficiency.

*5.1.2 Implementation Details.* In our implementation, the Neural Basis Extension network is deployed using tiny-cuda-nn ([29]) and consists of two hidden layers, each layer containing 64 neurons. The activation function is LeakyReLU, and the network outputs a 16-dimensional vector through the tanh function. We use the same loss function as the original 3D-GS [16]:

$$\mathcal{L} = (1 - \lambda) \cdot \mathcal{L}_1 + \lambda \cdot \mathcal{L}_{SSIM} \qquad (9)$$

The $\mathcal{L}_1$ loss computes the absolute error between the predicted image and the real image. $\mathcal{L}_{SSIM}$ represents the Structural Similarity Index, which measures the visual similarity between two images. We set $\lambda$ to 0.2 to strike a balance between pixel-level error and perceptual error in all of our experiments.

We keep the Gaussian parameter update strategy from 3D-GS [16] and set the learning rate of the Neural Basis Extension (NBE) to 0.001. We use the Adam optimizer to train the model for 30k iterations. To improve training stability, we initially use only spherical harmonics and introduce neural basis functions after 3k iterations to avoid abnormal jitter and overfitting problems in the early stages of training. In addition, we set the scaling factor $\alpha$ in the perturbation strategy to 0.3. The number of shared coefficients is set to be 16, which matches the spherical harmonics from 0 to 3rd degree. We set the prune ratio to be 60%, which is a suitable number to maintain the optimal performance for the most scene. While these settings are generally effective, fine-tuning may be necessary for a minority of models to achieve higher accuracy. All experiments were conducted on a dedicated workstation with an Intel Core i9-10900 CPU, 64GB of RAM, and an NVIDIA GeForce RTX 3090 GPU with 24GB VRAM.

*5.1.3 Bakery and Pruning Implementation.* We bake the NBE network output into 16 images of $400 \times 400$ resolution and modify the original 3D-GS rendering pipeline to load the baked data into memory and pass it to the GPU for real-time rendering. This modification supports network queries by the CUDA kernel without compromising performance. To compute the importance score of each 3D Gaussian, we extend the original diff-gaussian-rasterization module to capture the relevant indicators of each Gaussian while rendering the image. After pruning, we re-optimize the network and Gaussian parameters for 10k steps.

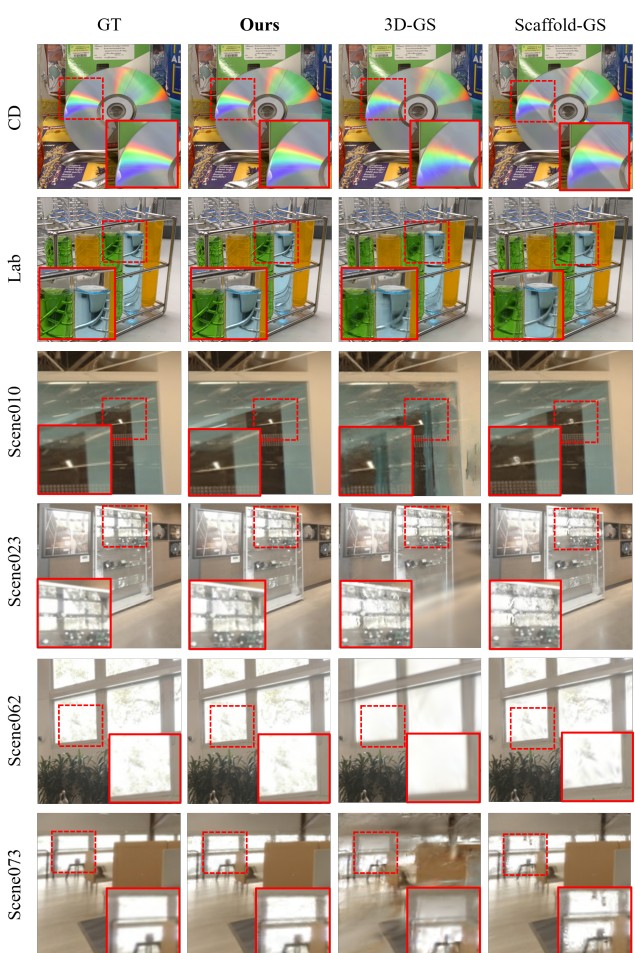

**Figure 4: Qualitative results of our method in complex scenes. Notice GSNB's performance on reconstruct rainbow ribbons on discs, light refraction in test tubes, and specular reflections on glass.**

## 5.2 Experimental Results

We compared GSNB to several state-of-the-art methods across various datasets. Our comparisons were made with the most relevant state-of-the-art methods, including 3D-GS [16], Scaffold-GS [25], and several NeRF-based methods such as NeX [41], Plenoxels [33], Instant-NGP [28], Mip-NeRF [1], Mip-NeRF360 [2], and Point-NeRF [44]. We color each cell as best , second best and third best .

*5.2.1 Results on Shiny and Spaces Datasets.* We report comparison results of our method against NeX [41], 3D-GS [16], and Scaffold-GS [25] on the Shiny and Spaces datasets in Table 1 and 2. We also show qualitative results in Figure 4. These datasets exhibit physical phenomena that challenge the original 3D-GS. We select four representative scenes from the Shiny dataset for evaluation. In the Spaces dataset, our method is assessed across eight scenes. Our method demonstrates significant improvements, particularly

**Table 1: Quantitative comparisons across 4 scenes in Shiny [41] dataset. Compared to the original model without pruning, our pruned model exhibits the highest rendering quality and the fastest speed at the same time.**

| Method \| Metric | PSNR↑ | SSIM↑ | LPIPS↓ | FPS↑ | MEM ↓ |
|---|---|---|---|---|---|
| NeX [41] | 29.00 | 0.939 | 0.143 | - | - |
| 3D-GS [16] | 28.51 | 0.913 | 0.128 | 211 | 320MB |
| Scaffold-GS [25] | 29.69 | 0.918 | 0.118 | 66 | 223MB |
| Ours | 30.13 | 0.925 | 0.115 | 224 | 266MB |
| Ours w/ pruning | 30.20 | 0.922 | 0.121 | 353 | 105MB |

**Table 2: Quantitative results across 8 scenes in Spaces [9] dataset. Our pruned model achieves the highest quality and rendering speed at the same time.**

| Method \| Metric | PSNR↑ | SSIM↑ | LPIPS↓ | FPS↑ | MEM ↓ |
|---|---|---|---|---|---|
| NeX [41] | 36.13 | 0.986 | 0.084 | - | - |
| 3D-GS [16] | 31.69 | 0.921 | 0.152 | 494 | 85MB |
| Scaffold-GS [25] | 37.41 | 0.973 | 0.046 | 126 | 76MB |
| Ours | 38.94 | 0.975 | 0.053 | 421 | 77MB |
| Ours w/ pruning | 40.63 | 0.980 | 0.051 | 550 | 32MB |

**Table 3: Quantitative results on the Mip-NeRF360 [2] and Tanks&Temples [17] datasets show the average model performance, underscoring the efficacy of our method across the majority of cases.**

| Dataset | Mip-NeRF360 | | | Tanks&Temples | | |
|---|---|---|---|---|---|---|
| Method \| Metric | PSNR↑ | SSIM↑ | LPIPS↓ | PSNR↑ | SSIM↑ | LPIPS↓ |
| Plenoxels [33] | 23.622 | 0.669 | 0.442 | 21.08 | 0.719 | 0.379 |
| Instant-NGP [28] | 26.751 | 0.751 | 0.298 | 21.92 | 0.745 | 0.305 |
| Mip-NeRF360 [2] | 29.087 | 0.842 | 0.209 | 22.22 | 0.759 | 0.257 |
| 3D-GS [16] | 29.085 | 0.869 | 0.183 | 23.14 | 0.841 | 0.183 |
| Scaffold-GS [25] | 29.217 | 0.864 | 0.198 | 23.96 | 0.853 | 0.177 |
| Ours | 29.362 | 0.870 | 0.182 | 23.97 | 0.848 | 0.176 |

in rendering the color and brightness of highlights. Although NeX shows better SSIM results, our method achieves superior performance in terms of PSNR and LPIPS metrics. Furthermore, we have significantly shortened the training time compared to NeX. Taking the CD dataset as an example, we reduced the training duration from the 18 hours required by NeX to under 30 minutes on a single 3090 GPU.

*5.2.2 Results on Mip-NeRF360 Datasets.* The Mip-NeRF360 dataset contains a collection of panoramic images that capture a variety of scenes and lighting conditions with detailed 360-degree views. Table 3 shows our method's average performance on this dataset, outperforming previous approaches such as Mip-NeRF360 [2], 3D-GS [16], and Scaffold-GS [25] on all metrics. As shown in the Figure 5, our method significantly improves highlight detail.

*5.2.3 Results on Synthetic Datasets.* The Synthetic NeRF dataset is a widely used synthetic dataset. Table 4 shows the stable rendering

**Table 4: Quantitative results for 8 scenes in Synthetic NeRF dataset. We use the metric results provided by original papers except for 3D-GS, which were obtained in our own experiments.**

| Method \| Scene | Mic | Chair | Ship | Materials | Lego | Drums | Ficus | Hotdog | Avg. |
|---|---|---|---|---|---|---|---|---|---|
| Plenoxels [33] | 33.26 | 33.98 | 29.62 | 29.14 | 34.10 | 25.35 | 31.83 | 36.81 | 31.76 |
| Instant-NGP [28] | 36.22 | 35.00 | 31.10 | 29.78 | 36.39 | 26.02 | 33.51 | 37.40 | 33.18 |
| Mip-NeRF [1] | 36.51 | 35.14 | 30.41 | 30.71 | 35.70 | 25.48 | 33.29 | 37.48 | 33.09 |
| Point-NeRF [44] | 35.95 | 35.40 | 30.97 | 29.61 | 35.04 | 26.06 | 36.13 | 37.30 | 33.30 |
| 3D-GS [16] | 36.67 | 35.52 | 31.67 | 30.49 | 36.08 | 26.28 | 35.49 | 38.10 | 33.79 |
| Scaffold-GS [25] | 37.25 | 35.28 | 31.17 | 30.65 | 35.69 | 26.44 | 35.21 | 37.73 | 33.68 |
| Ours | 36.89 | 35.68 | 31.81 | 30.65 | 36.40 | 26.68 | 35.66 | 38.08 | 33.98 |

quality of our method. However, synthetic scenes lack the complex lighting conditions of the real world, in which case our model cannot fully demonstrate its power.

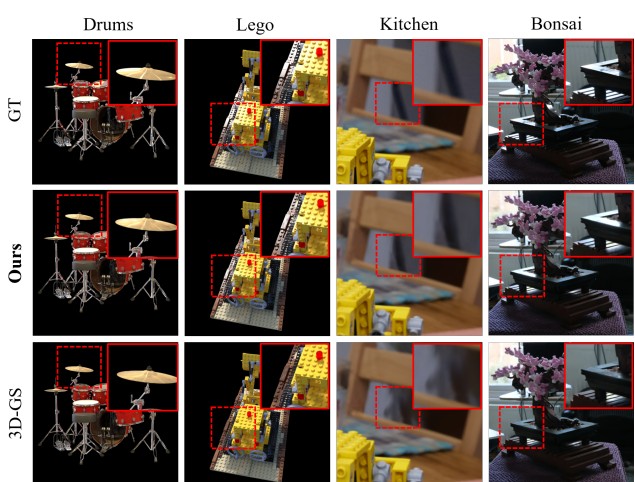

**Figure 5: Qualitative Results on Synthetic [27] and &Mip-NeRF360 [2] datasets**

*5.2.4 Results on Tanks and Temples Datasets.* Table 3 shows the performance of various methods on this dataset. We select the two scenes Truck and Train used in the original 3D-GS [16] for experiments. While Scaffold-GS [25] uses anchor points to deeply reform the original 3D-GS, our approach integrates a lightweight neural network to achieve comparable results.

*5.2.5 Results on Pruning & Re-optimization.* We evaluated our pruning approach on typical challenging scenes, such as CD and Lab, characterized by rainbow light bands and refraction phenomena. The results indicate that our method can effectively prune redundant Gaussians while maintaining the performance of both the original 3D-GS method and the enhanced GSNB. However, as the 3D Gaussian distribution becomes excessively sparse, there is an observable drop in performance. The Figure 7 illustrates that GSNB maintains the performance comparable to the converged 3D-GS model even using only 10% of the original model parameters. Notably, while we use the prune ratio for comparison, our method requires fewer Gaussians at the same ratio, thanks to the enhanced expressiveness provided by the Neural Basis Extension.

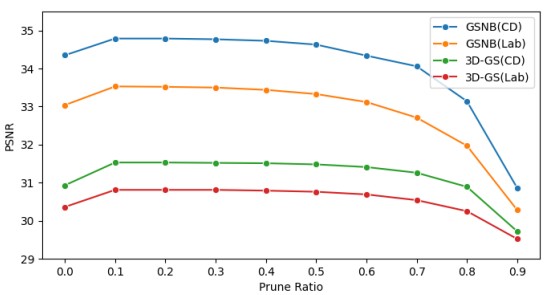

**Figure 6: Influence of the Pruning ratio on the performance of the pruned model after the re-optimization process, our strategy shows resistance to the reduction of Gaussians in both GSNB and original 3D-GS model.**

**Table 5: Quantitative results on Baking Operation. Compared to computing the network in real-time, our model with baked images achieve higher speed without hindering the rendering quality.**

| Scene | CD | | | | Lab | | | |
|---|---|---|---|---|---|---|---|---|
| Method \| Metric | PSNR↑ | SSIM↑ | LPIPS↓ | FPS↑ | PSNR↑ | SSIM↑ | LPIPS↓ | FPS↑ |
| 3D-GS [16] | 30.93 | 0.939 | 0.118 | 239.2 | 30.36 | 0.927 | 0.139 | 256.41 |
| Ours w/o baking | 34.35 | 0.954 | 0.098 | 27.21 | 33.04 | 0.952 | 0.110 | 32.92 |
| Ours | 34.42 | 0.955 | 0.1 | 294.12 | 33.05 | 0.953 | 0.112 | 253.16 |
| Scene | Tools | | | | Giants | | | |
| Method \| Metric | PSNR↑ | SSIM↑ | LPIPS↓ | FPS↑ | PSNR↑ | SSIM↑ | LPIPS↓ | FPS↑ |
| 3D-GS [16] | 27.68 | 0.921 | 0.153 | 75.76 | 25.09 | 0.866 | 0.103 | 58.69 |
| Ours w/o baking | 27.78 | 0.924 | 0.148 | 10.41 | 25.35 | 0.868 | 0.105 | 6.12 |
| Ours | 27.78 | 0.924 | 0.15 | 76.92 | 25.38 | 0.868 | 0.106 | 56.82 |

**Table 6: Quantitative results on Perturbation Strategy**

| Scene | Mic | Chair | Ship | Materials | Lego | Drums | Ficus | Hotdog | Avg |
|---|---|---|---|---|---|---|---|---|---|
| Ours w/o perturbation | 36.87 | 35.42 | 31.69 | 30.58 | 36.41 | 26.27 | 35.44 | 37.94 | 33.82 |
| Ours | 36.89 | 35.68 | 31.81 | 30.65 | 36.40 | 26.68 | 35.66 | 38.08 | 33.98 |

## 5.3 Ablation Studies

We conducted ablation experiments on baking operations and perturbation strategies. Our results show that baking can greatly improve the rendering speed of the model while ensuring the rendering quality. And the perturbation strategy can effectively improve the generalization ability of GSNB in simple scenes to avoid overfitting.

*5.3.1 Baking Operation.* Baking plays an important role in achieving the real-time rendering performance of GSNB. We test the baking performance in some of the most challenging scenarios. Experimental results in Table 5 show that the impact of baking on rendering quality is almost negligible. Compared to real-time neural network inference, this strategy significantly speeds up the rendering.

*5.3.2 Perturbation Strategy.* To investigate whether the perturbation strategy can increase the generalization ability of the model, we conduct experiments on the Nerf synthetic dataset that is prone to overfitting. The perturbation value is uniformly set to 0.3 to

evaluate the perturbation strategy. Experimental results in Table 6 show that the strategy achieves certain progress in most scenarios.

*5.3.3 Sensitivity Analysis of Spherical Harmonic Degree.* To investigate the effect of the degree of spherical harmonics, we test spherical harmonics from 0 to 3 degree on the Lego and Drums datasets. As shown in Figure 7, the results indicate that the spherical harmonics primarily use the 0 degree part of the sharing coefficient and the degree of spherical harmonics has little impact on the overall performance of the model. We recommend using higher degree for simpler models and vice versa.

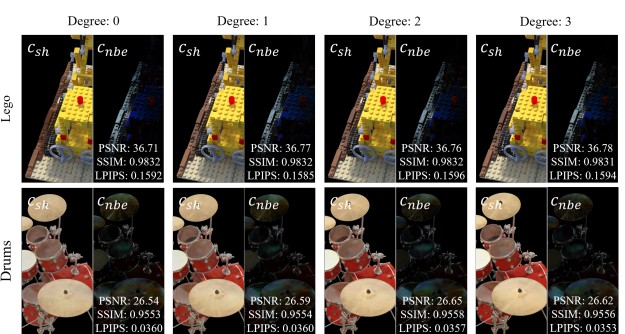

**Figure 7: The impact of different degrees of spherical harmonics on our method. Despite the varying degrees, the Neural Basis Extension consistently captures high-frequency information.**

## 6 CONCLUSION

We present Gaussian Splatting with Neural Basis Extension (GSNB), a method that extends the capabilities of 3D-GS in rendering complex visual environments while reducing memory requirements. Using a hybrid appearance model, GSNB supplements the color computation of 3D-GS and amplify the fidelity of visual effects. This integration is further strengthened by the incorporation of a set of regularization methods to enhance the robustness of our model in complex scenes. Our studies on both challenging dataset and general dataset demonstrate GSNB's ability to provide high quality rendering while maintaining optimal efficiency.

Meanwhile, our method inherits some inherent limitations from 3D-GS, particularly in geometric accuracy, as the use of neural networks for modeling view-dependent effects can lead to inaccuracies in capturing the inherent geometry of scenes. Future research could explore incorporating geometric constraints such as surface normals and depth maps into the training process of Neural Basis Extension, guiding the neural network towards more precise scene reconstruction.

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
