# OpenReview forum: "Gaussian Splatting With Neural Basis Extension"
_acmmm.org/ACMMM/2024/Conference — MM2024 Poster_

### Official Review · Reviewer_quLi · 2024-05-23

**Rating:** 5
**Confidence:** 3

**Summary:**

The paper aims to improve the 3D Gaussian Splatting (GS) method from two perspectives: enhancing expressiveness limited by spherical harmonics and achieving more compact representations. To tackle these issues, the authors propose three key things:

- Using a lightweight MLP, named Neural Basis Extension (NBE), to augment spherical harmonics, resulting in markedly enhanced rendering quality.
- Baking NBE into images achieves the benefits of using NBE while reducing its costs to a negligible level.
- Introducing a novel importance score for pruning.

**Strengths:**

The idea of using a baked, image-like structure to express higher color expressibility without significant computational and memory overhead is innovative and practical.
The proposed method demonstrates significant improvement across various datasets, even with the reduced size, which are challenging to represent using spherical-harmonics-only GS.
Although only mentioned in the appendix, the comparison with various pruning strategies is noteworthy and provides valuable tips for using GS-based rendering.

**Limitations:**

- Neural Basis Extension

A question arises regarding the necessity of the network (Neural Basis Extension, NBE) if the baked representations are sufficient for higher expressiveness.
If the network is ultimately baked into an image-like structure, it may be beneficial to start directly from the table (multi-channel images) and optimize only that.
If there are no significant problems with directly optimizing the table, it might be better to discard the network and focus solely on the image-like structures.

- Novelty of Importance Scores

The proposed importance score is similar to those that are already presented by RadSplat and LightGaussian.

- Pruning Strategies

Soft filter pruning is only mentioned in the related works, yet experiments in the appendix demonstrate its effectiveness. These results should be moved to the main paper to provide a more comprehensive evaluation of the proposed method and to highlight the practical benefits of the pruning strategy.

**Suitability:**

2

---

### Official Review · Reviewer_qSdf · 2024-05-24

**Rating:** 4
**Confidence:** 3

**Summary:**

The author proposes a novel approach called GSNB that represents the appearance model as the combination of a lightweight MLP and common spherical harmonics. Along with that, two methods that aim to accelerate the rendering process are proposed: baking and pruning, both contribute to efficiency. The method shows significant improvements in several datasets.

**Strengths:**

Clear written and organized,

Clearly better quantitative results against relevant baselines, with good ablation studies.

The paper aims at an important topic for immersive virtual reality.

**Limitations:**

- [Novelty]

In terms of the paper's novelty, I believe the main contribution is the neural basis function you proposed. However, in the related work section, I have not seen any other studies with the same purpose as yours, so you should provide a more detailed comparison with existing work in related works section. In fact, the Scaffold-GS you included in the teaser figure also uses an MLP to model color; Moreover, the baseline NeX (NeX: Real-time View Synthesis with Neural Basis Expansion) also incorporate the concept of neural basis expansion. It would be better if you compare the differences between your work and above mentioned efforts.

- [Motivation of Annealing Perturbation Against Overfitting]

It’s not easy to follow the motivation behind the design in Section 4.3. You explain at the beginning of Section 4.3 that the color direction for each Gaussian is also fixed, but I don't quite understand this because the position of the GS is also changing. I am more inclined to interpret Section 4.3 as a training trick, and I feel that the reasons behind this trick are not clearly explained. Of course, this trick did play a certain role. I hope you can provide a clearer explanation of the real reasons behind it. This makes your paper more logically rigorous.

- [Some mistakes]

The paper is generally well-composed, but there are a few minor typographical errors that need attention.

On line 435, the word "halve" should be corrected to "have."

In line 806, the reference should be to Figure 6, not Figure 7.

**Suitability:**

2

---

### Official Review · Reviewer_ZvAQ · 2024-05-24

**Rating:** 5
**Confidence:** 3

**Summary:**

This paper aims to enhance the ability of 3D-GS to model view-dependent color while reduce space consumption. The authors propose Gaussian Splatting with Neural Basis Extension with three contributions:
1) a hybrid appearance model that utilizes a lightweight MLP to extend the color expressed by spherical harmonics,
2) an efficient baking method,
3) and a new Gaussian pruning strategy that speeds up the rendering process.
The proposed method is validated on several standard testset, and it achieves good rendering quality with fast rendering speed.

**Strengths:**

1. The four modifications for 3D-GS(corresponding to 4.2-4.5) are all reasonable, each with a clear purpose.
2. The results are impressive. The ablation studies are also high quality and clearly show improvements for rendering quality.
3. The paper is well organized and easy to follow.

**Limitations:**

1. Related works and comparision
Please add works related to the modeling of high-frequency information or reflective phenomena in the Related Work section, and discuss the distinctions between the proposed work and the previous works. For instance, regarding NeX and Scaffold-GS mentioned in comparison experiments.

2. Discussions
I am wondering why not directly use higher-order spherical harmonics to model view-dependent color for enhancing effects like specular highlights and light refraction? Is there any inherent trade-off? It would be better to highlight the difference between “higher-order spherical harmonics” and “lower-order spherical harmonics + neural basis function”.

3. Minor typo error
The manuscript is well written, except some details:
Line 435: “halve” -> “have”
Line 806: should be referred to Figure 6, instead of Figure 7

**Suitability:**

2

---

### Official Review · Reviewer_mbLu · 2024-05-24

**Rating:** 2
**Confidence:** 3

**Summary:**

The paper addresses the limitations of Gaussian fields that rely on simple spherical harmonics and a splitting strategy that increases memory usage and slows down rendering. It incorporates a Neural Basis Extension (NBE) into 3DGS, specifically using a lightweight MLP to share feature coefficients with Spherical Harmonics. This approach achieves better results in lighting and reflection conditions. Additionally, the authors use a baking technique to precompute the MLP and devise an importance score for model pruning, achieving high-quality and real-time rendering.

**Strengths:**

1. The paper is well-organized, with informative figures that effectively illustrate the concepts discussed.
2. The qualitative results clearly demonstrate the efficacy of the proposed method.
3. The proposed method is evaluated on multiple datasets and supported by both quantitative and qualitative results.

**Limitations:**

1. Despite addressing an important problem, the paper lacks clarity in some sections and there are multiple steps in the methodology that are not clearly defined or introduced in the manuscript, particularly regarding the baking and precomputation processes. The code is not released, leaving the implementation details unclear.

2. The impact of the NBE, baking, and pruning processes on training time is not adequately discussed, and these parts might have compromised the training duration.

3. Although there are abundant quantitative results, the paper needs more qualitative results, especially for large-scale outdoor scenes (e.g., the Train dataset in T&T) where lighting and reflection conditions vary significantly from indoor scenes, which could better showcase the efficacy of NBE.

4. The paper should include both quantitative and qualitative comparisons with SOTA 3DGS compression methods, such as Compressed 3dgs [1]. Presenting comparisons of compression ratios will make the model convincing.
[1] Niedermayr, S., Stumpfegger, J., & Westermann, R. (2023). Compressed 3d gaussian splatting for accelerated novel view synthesis. arXiv preprint arXiv:2401.02436.

5. Moreover, the paper should incorporate related work on 3DGS pruning method.

6. There is a lack of ablation studies demonstrating how the NBE works.

**Suitability:**

3

---

### Meta-Review · Area_Chair_vbMP · 2024-06-28

**Recommendation:** Accept (Poster)
**Confidence:** 4

**Metareview:**

This paper originally received a set of mixed reviews. The rebuttal successfully convinced all the reviewers. Therefore, we are happy to recommend the acceptance of this paper.